

# **The referential grain size and effective porosity in the Kozeny-Carman model**

K. Urumović[1] and K Urumović Sr[2]
[1] Croatian Geological Survey, Sachsova 2, P.O. box 268, HR-10001 Zagreb, Croatia
[2] Ulica Lea Müllera 3. odvojak 2, 10090 Zagreb, Croatia
Correspondence to: Kosta Urumović, kosta.urumovic@hgi-cgs.hr



# Abstract

In this paper, the results of permeability and specific surface area analyses as functions of granulometric composition of various sediments (from silty clays to very well-graded gravels) are presented. The effective porosity and the referential grain size are presented as fundamental granulometric parameters expressing an effect of the forces operating on fluid movement through the saturated porous media. This paper suggests procedures for calculating referential grain size and determining effective (flow) porosity, which result in parameters that reliably determine the specific surface area and permeability. These procedures ensure the successful application of the Kozeny-Carman model up to the limits of validity of Darcy's law. The value of effective porosity in the referential mean grain size function was calibrated within the range of 1.5 μm to 6.0 mm. The reliability of the parameters applied in the KC model was confirmed by a very high correlation between the predicted and tested hydraulic conductivity values ($R^2$=0.99 for sandy and gravelly materials; $R^2$=0.70 for clayey-silty materials). The group representation of hydraulic conductivity (ranging from $10^{-12}$ m/s up to $10^{-2}$ m/s) presents a coefficient of correlation of $R^2$=0.97 for a total of 175 samples of various deposits. These results present new developments in the research of the effective porosity, the permeability and the specific surface area distributions of porous materials. This is important because these three parameters are critical conditions for successful groundwater flow modeling and contaminant transport. Additionally, from a practical viewpoint, it is very important to identify these parameters swiftly and very accurately.



# 1    Introduction

The effect of the granulometric composition of granular porous media on its transmissivity, accumulation and suction parameters is both a permanent scientific challenge and a practical issue. In hydrogeology, particular attention is given to hydraulic conductivity. Hazen (1892) and Slichter (1902) have published widely accepted and reputable models for calculating the hydraulic conductivity of uniform sands using effective grain size. The term "effective grain", used for grain diameters in both formulae could lead to confusion (Mavis and Wilsey, (1936). However, Hazen's formula uses $D_{10}$ (soil particle diameter where 10% of all soil particles are finer (smaller) by weight), and Slichter proposes using the mean diameter. This confusion persisted, and in recent decades, grain size $D_{10}$ has been misused frequently (Kovács 1981), (Vukovic and Soro 1992), (Cheng and Chen 2007), (Odong 2008) in formulae that actually use another effective grain size.

The usage of certain forms of mean grain size became inevitable with the development of hydraulic conductivity models that describe relations between the hydraulic conductivity and the specific surface area (Krüger 1918), (Zunker 1920), (Blake 1922), (Kozeny 1927) (Fair i Hatch 1933). (Kozeny 1927) introduced the equation of permeability for the flow model containing a bundle of capillary tubes of even length. Kozeny's permeability formula was later modified by (Carman 1937) and (Carman 1939). Carman redefined specific surface area and presented it as a conversion of mean grain size and the index of porosity and incorporated an effect of tortuosity for the flow around individual grains. The resultant form of the equation is known as the Kozeny-Carman's (KC) equation. The verity of the KC formula application results is strongly dependent on the verity of effective porosity and representative grain size. (Kozeny 1927) used the harmonic mean grain size of samples. (Bear 1972) recommended the same grain size. (Koltermann i Gorelick 1995) and (Kamann, et al. 2007) stated that the harmonic mean performed best in samples with high fine grain contents. Chapuis and Aubertin (2003) proposed laboratory tests for determining the specific surface area of fine grained materials for application in the KC formula.

The objective of this article is to research the relationship between average mean grain size and effective porosity in relation to permeability and specific surface area for a wide range of grain sizes and particle uniformities in various soil samples. In the hydraulic conductivity calculations, the Kozeny-Carman equation was used to discover the algorithm for calculating the referential mean grain size. This grain size, along with effective porosity, generates a harmonious parametric concept of the impact of porous media geometrics on its transmission capacity.

# 2    Study area and analyzed deposits

For the purpose of this work, data on sandy and gravely aquifers and clayey-silty deposits were collected. All of the study sites are located in the plains of the Republic of Croatia (Fig. (1)). The northern parts of the Republic of Croatia are covered by thick quaternary deposits with sandy and gravely aquifers (Brkić et al. 2010). Covering aquitards are composed of silty-clayey deposits.

Figure 1. The map of Northern Croatia with test sites locations





The analyses of non-cohesive deposits were conducted on 36 gravel test samples from six
investigation boreholes on the Đurđevac well field (marked as GW on Fig. (1); 19 uniform sand test
samples from the investigation boreholes on two well fields – Beli Manastir (marked as SU1) and
Donji Miholjac (marked as SU2); and 28 samples of sand with laminas made of silty material from
two investigation boreholes on two well fields – Ravnik (marked as FS/SU1) and Osijek (marked as
FS/SU2). Appropriate pumping tests were conducted on the test fields to determine the average
hydraulic value of aquifers.
Cohesive deposits were investigated on three sites. Soil samples from exploration boreholes
(depth 1.0 – 30.0 m) were laboratory tested. Analyses on granulometric composition (grain size
distribution), hydraulic conductivity and Atterberg limits were conducted. On the first test field (route
of Danube, Sava channel; marked as CI/MI1), all the aforementioned analyses were conducted for
each soil sample. Sixty-five samples of various soil types were analyzed. On the second and third test
sites (Ilok, marked as CI/MI2, and Našice, marked as CI/MI3), loess and aquatic loess-like sediments
were investigated. Laboratory analyses were conducted on 21 samples from eight investigation
boreholes. Specific analyses at various depths were conducted on the samples from this test site, and
on account of this, the mean values for the individual boreholes were correlated (K. Urumović 2013).

## 3  Methodology

### 3.1  Hydraulic model

The effects of porosity $n$ and specific surface area $a$ on fluid movements in porous media can
be illustrated by analyzing the force field in the representative elementary volume (REV) $\delta V = \delta A \delta s$
(Fig. (2)) in the direction of elementary length $\delta s$ that is perpendicular to the elementary plane $\delta A$.
Figure 2. Definition sketch of liquid driving and opposed viscous forces for elemental volume
The forces of pressure and gravity cause the motion of the fluid in the pores. A pressure force
is transferred to $\delta s$ between the entry plane $\delta A$ and its parallel exit plane. The total amount is
proportional to the gradient $\delta p/\delta s$. A component of the gravity force $\rho g$ in the fluid volume $n\delta A \delta s$ is
proportional to the sine of the angle made by $\delta s$ with its projection on the horizontal plane. This equals
$\rho g n \delta A\, \delta s\, \partial z / \partial s$. These two driving forces are, in fluid motion, against the force of viscosity $\tau$. The
force of viscosity is proportional to the viscosity coefficient of water $\mu$, the average velocity $q_s$ of
water flow in direction $\delta s$, and the effect of the geometry of void space, which is given by the drag
resistance constant $r_s$ in direction $\delta s$ and is proportional to the specific surface area. When the water
flows, these forces are in balance, and hence (Hantush 1964), (S. K. Urumović 2003):

$$-n\delta V \frac{\partial p}{\partial s} - n\delta V \rho g \frac{\partial z}{\partial s} - \delta V \mu r_s q_s = 0 \qquad (1)$$

or:

$$q_s = -\frac{n\rho g}{r_s \mu}\frac{\partial (p/\rho g + z)}{\partial s} = -\frac{n\rho g}{r_s \mu}\frac{\partial h}{\partial s} = -K_s \frac{\partial h}{\partial s} = -k_s \frac{\rho g}{\mu}\frac{\partial h}{\partial s} \qquad (2)$$

These relations express Darcy's law, as theoretically described by Hubbert (1956). Here, the focus is
on permeability as a property of porous media that is (in Eq. (2)) given by the relation $k_s = n/r_s$, $k_s$ [L$^2$].





Porosity $n$ is measured as the volume of moving fluid and is connected with the specific effect of the
driving forces of pressure and gravity. The constant $r_s$ expresses an effect of void geometry on the
amount of viscosity forces and represents the extent of the effect of void geometry on water retention.
The size of this effect is equivalent to a specific surface area $a_p$, [L$^{-1}$] inside the porous media, that is,
to a relation between 1) the surface of the solid grains that confronts the water flow and 2) the
saturated void volume that transfers the flow driving force. Following the Hagen Poiseulle law, the
specific surface area $a_p$ [L$^{-1}$] is inversely proportional to the hydraulic radius $R_H$ [L]. Thus, in an
isotropic environment, $r_s \propto a_p^2$, the permeability is given as follows:

$$k = \frac{n}{r_s} = C\frac{n}{a_p^2} = CnR_H^2 \qquad (3)$$

where $C$ represents the dimensionless coefficient of proportionality that is dependent on the particle
shape. $R_H = 1/a_p$ represents the hypothetical hydraulic radius of the porous media and the impact of the
specific surface area of effective flow voids (Irmay 1954).

## 3.2 Geometric parameters of permeability

There are four ways to express the specific surface area $A_s$ [L$^2$] based on solid volume, V$_s$[L$^3$].
They are as follows:
$a_p$ [L$^{-1}$] – specific surface area based on the volume of contented pores $V_p$;
$a_T$ [L$^{-1}$] – specific surface area based on the total volume (solids + pores) $V_T$;
$a_m$ [L$^2$M$^{-1}$] – specific surface based on the mass of solids $M_s$;
$a_s$ [L$^{-1}$] – specific surface area based on the volume of solids $V_s$ of density $\rho_s$
All of the above-mentioned forms of specific surface area are related to the hydraulic radius of porous
media $R_H$. The relationship between these forms is given by the following expression:

$$a_p = \frac{A_s}{V_p} = \frac{a_T}{n} = \frac{\rho_s(1-n)}{n}a_m = \frac{(1-n)}{n}a_s = \frac{1}{R_H}. \qquad (4)$$

Kozeny (1927) used Eq. (4) with $a_T$. He developed a theory for a bundle of capillary tubes of equal
length. Carman (1937) verified the Kozeny equation and expressed the specific surface per unit mass
of solid as $a_m = A_s/M_s$, such that it does not vary with porosity. Furthermore, Carman (1939) tried to
consider the tortuosity of the porous media by introducing an angular deviation of 45° from the mean
straight trajectory. He obtained the best fit from the experimental results with a factor C=0,2 in Eq.
26 (3).

In hydrogeology, the specific surface area is often presented with a conversion of mean grain
diameter $D_m$. Permeability is given by the following expression (Bear 1972):

$$k = \frac{n^3}{180(1-n)^2}D_m^2 \qquad (5)$$

This relation has been achieved by inserting the solid specific surface area ($a_s = 6/D_m$) from Eq. (4) into
Eq. (3) with $C=0,2$. This solution of the Kozeny-Carman equation (Bear 1972) is given for uniform
sphere particles. Thus, the critical factors of porous media transmissivity are effective porosity $n$ (in
the form of porosity function) and referential mean grain diameter $D_m$. Grouping these terms
functionally gives the following expression:

$$K = C\frac{n_e}{a_p^2} = \frac{n_e}{180}\left(\frac{n_e}{(1-n_e)}D_m\right)^2 \qquad (6)$$





Figure 3. Effects of driving (n) and drag resistance (n²/(1-n)²) factors in porosity function (n³/(1-n)²)

4       Evidently, the effective porosity $n_e$, has a direct impact on the magnitude of driving forces and

an indirect impact as $n_e^2/(1-n_e)^2$ (Fig. 3) on the conversion of the specific surface value into a value of
the referential mean grain diameter, which is the carrier of drag resistance. Both of the aforementioned
forces affect the moving fluid. Therefore, effective porosity is an active factor only in relation to the
pores through which the water flows.

## 9   3.3   Referential grain size

10       Many authors present the Kozeny-Carman equation with $D_m^2$ instead of $a_s^2$ in Eq. (5) without

completely indicating the calculation of this equivalent mean diameter. In engineering practice, there
are three ways to calculate the mean of the rated size of adjacent sieves:
Arithmetic:       $d_{i,a}=(d_{i<}+d_{i>})/2$           (7)
Geometric:       $d_{i,g}=\sqrt{d_{i<}\times d_{i>}}$           (8)
Harmonic:       $d_{i,h}=2/[(1/d_{i<})+(1/d_{i>})]$     (9)
where $d_{i<}$ [L] is the smallest grain and $d_{i>}$ [L] is the largest grain in the segment. It can be shown that
$d_{i,h}<d_{i,g}<d_{i,a}$, across all cases. However, the difference is not significant. Todd (1959) recommends the
use of the geometric mean. Bear (1972) prefers the harmonic mean. Recent authors often follow these
recommendations.

20       The integration of all of the mentioned grain sizes (Eq(s) (7), (8), (9)) in the sieve residue

across the entire sample has a crucial effect on the mean grain size value. An overview of both the
related expert and scientific literature indicates the use of either the arithmetic mean:

$$D_a=\frac{\sum P_i d_{i,a}}{100}$$       (10)

or the harmonic mean:

$$D_h=\frac{100}{\sum(P_i/D_{i,h})}$$       (11)

which is the sum of mean grain sizes in sieve residue $d_i$. Here, $P_i$ is a percentile of the sieve residue
mass in the total mass of the sample. Accurate results of permeability and specific surface were only
achieved for the uniform deposits of sand and silt (Chapuis and Aubertin 2003), (Kasenow 1997).
Major errors resulted from applying Eqs. (10, 11) for samples with a wide range of particle sizes.
Similar observations were noted in sedimentology and soil science research. Arkin and Colton (1956)
noted that the arithmetic mean may be significantly distorted by extreme values and therefore may not
be appropriate. For soil samples, Irani and Callis (1963) advocated the use of geometric rather than
arithmetic statistical properties. The reason, in part, is that in a natural soil sample there is wide range
of particle sizes making the geometrical scale much more suitable then the arithmetic scale. The
general mathematical expressions for calculating the geometric particle size diameter $D_g$ of the sample
are as follows:

$$D_g=EXP\left[\frac{1}{M_s}\sum m_i\ln(d_{i,g})\right]$$       (12)

or

$$D_g=EXP\left[0,01\sum P_i\ln(d_{i,g})\right]$$       (13)





where $M$ [M] represents the mass of the sample and $m_i$ [M] represents the mass of particular sieve
residues, $P_i = 100m_i/M$. It can be shown that $D_h < D_g < D_a$. This difference is very small when calculated
for uniform deposits but rapidly grows when calculated for the mean grain sizes of poorly sorted
deposits. In the case of gravelly sediments, the difference may reach up to 2 orders of magnitude.



## 3.4 Porosity factor

In a permeability model, the porosity function expressed by porous media transmissivity factors (Eq. (6)) applies only to flow pores (Eq. (2)). Accordingly, it was named effective porosity. The effective porosity could sometimes differ from the specific yield, which is a drainable porosity, determined in a laboratory. The numerical difference between the effective porosity and the specific yield may not be discernible when analyzing uniform sand, but it can increase significantly when analyzing samples containing a greater percentage of small size (clay, silt) particles. Expressions of specific yield functions of granulometric aggregates (Eckis 1934) or median grain size (Davis and De Wiest 1966) are unsuitable in permeability equations (Eq. (6)) for two reasons. First, in these figures, specific yield was not shown in relation to referential grain size ($D_g$). Second, the specific yield represents the drainage in negative pressure conditions. Effective porosity represents the active pores at the time of fluid flow for a sample of certain $D_g$, as shown in this paper. These relations were based on the analysis of data from several samples of various deposits (from clay to gravel). The initial values of porosity used in this procedure were ranges of an average specific yield value (Fig. (4)), according to the data from the U.S. Geol. Survey Water Supply Paper (Morris and Johnson 1967). The laboratory reputation and a large number of analyses (33 samples of gravel, 287 of sand and 266 of silt and clay) provided a high quality base for the identification of the mean value of a specific yield range.

Figure 4. Range and arithmetic mean of the specific yield values for 586 analyses in Hydrol. Lab. of the U.S. Geol. Survey (from Morris & Johnson, 1967)

The value of effective porosity is slightly lower than the value of the specific yield. This value is related to the referential mean grain size ($D_g$), forming the function of drag resistance effect in the water flow through a porous media (Eq. (6), Fig. (3)). The reliable reconstruction of the effective porosity range (Fig. (5)) was ensured through the strong impact of the discussed form of the porosity function ($n^3/(n-1)^2$) (Fig. (3)) and the accurate calculation of referential mean grain size (Eq. (12), Eq. (13)). These relations simultaneously verified the applicability of the Kozeny-Carman equation for a wide range of granulometric composition, in terms of both grain size (samples with $D_g$ from 1.5 µm up to 6 mm) and grade (Fig 5).

Figure 5. Relation between referential mean grain $D_g$ and effective porosity $n_e$. Note: Dot line divides uniform grain deposits U=D60/D10<2, and medium uniform grain deposit 2<U<20. Verified samples of non-uniform grain deposits of sand and gravel (U>20) lie below the full line

## 4 Results and verification

Reliable verification of the analyzed parameter relations for a wide range of granulometric compositions was conducted using the Kozeny-Carman equation and the analyses of the hydraulic conductivity researched deposits in situ as well as in the laboratory. Hydraulic conductivity K [LT$^{-1}$] given by the KC equation (according to Eq. (6)) is:

$$K=\frac{\rho g}{\mu}\frac{n_e^3}{180(1-n_e)^2}D_m^2=0{,}0625D_g^2\frac{n_e^3}{(1-n_e)^2} \qquad (14)$$





where $\rho$ [ML$^{-3}$] represents the density and $\mu$ [ML$^{-1}$T$^{-1}$] represents the viscosity of water, with gravity $g$
[LT$^{-2}$]. The coefficient 0.0625 is correct for a diameter of the referential mean grain $D_g$ expressed in
mm and a water temperature of 10°C. Hazen's (1892) non-dimensional temperature correction factor
$\tau=0.70+0.03T$ ($T$ - temperature in °C) was used to present an effect of temperature difference,
ensuring an error less than 2% for T<30°C.
The Kozeny-Carman equation is actually a special form of Darcy's law (in the case of the unit
value of hydraulic gradient). Hence, it should be applicable across all possible natural samples of
porous media. The hydraulic testing of natural deposits poses a problem in correlation investigations.
Non-cohesive deposits make it almost impossible to ensure the laboratory testing of the content and
distribution of particles or to consolidate material in its natural and undisturbed state. The average
hydraulic conductivity calculated by analyzing the pumping test data was used for correlation in the
non-cohesive deposits. Test sites were chosen to fulfill the following criteria: the borehole core must
be of a 100% natural lithological compound, and the analysis of particle size distribution must be
conducted on the core samples. If the exploration borehole was located in the vicinity of the tested
well, the hydraulic conductivity of the local scale was used. If there were more boreholes at a greater
distance from the pumped well, the hydraulic conductivity of a sub-regional scale was determined and
used for correlation. Values of the predicted K appropriate to the test data scale, obtained from the
grain size distribution analysis, were averaged. Silty and clayey samples were processed in a specific
way. If a specific sample was analyzed in the laboratory (grain size analysis and hydraulic
conductivity), the results were (both literally and functionally) on a laboratory scale.
The criteria for evaluating the acceptable accuracy of the predicted hydraulic conductivity,
expressed by its correlation with a tested K value, should not be equal for different types of materials.
Chapuis and Aubertin (2003) of the *École Polytechnique de Montréal* conducted a very interesting
study. They concluded that the acceptable accuracy of a predicted value of K for clayey materials is
between 1/3 and 3 times the measured K-value, which is within the expected margin of variation for
the laboratory permeability test. That relation is referred to a calculation of K by the Kozeny-Carman
equation using a specific surface area determined in the laboratory. Such criteria can definitely be an
acceptable accuracy limit for calculating the K using referential grain size. In the case of silty, non-
plastic soils, three specimens of the same sample may give $K$-values ranging between ½ and 2 times
the mean value. An excellent precision (K-value within ±20%) can be reached with sand and gravel
when the special procedure is applied (Chapuis and Aubertin 2003). These criteria were accepted for
hydraulic conductivity calculations using the KC equation and applying the effective porosity and
referential mean grain size. The accepted criteria require a high level of accuracy for determining the
referential mean grain size and effective porosity in their roles in Eq. (14).
In the verification process, the results acquired using the KC equation were matched with the
results of the hydraulic tests. The average local K-values of sandy aquifers were identified (pumping
test data) and compared to the average sample K value. Verification of K-values for the gravelly
aquifer is of a sub-regional scale because the boreholes that provided the high-quality core were
located at a distance of 150 – 500 m from the pumped well. The tested value of hydraulic conductivity
was determined by analyzing a series of successive steady states. The third case was of a laboratory
scale where K-values of cohesive materials were analyzed. The hydraulic conductivity values of silty-
clayey samples and the granulometric parameters were the results of the laboratory testing of each
sample. The criteria for correlating predicted and tested K-values were customized to these
procedures.


## 4.1 Incohesive deposit

The results of the calculation of hydraulic conductivity using the KC formula (Eq 14) for individual samples of sand and gravel were presented graphically, according to borehole depths. The average values of hydraulic conductivity for individual pilot fields are presented in the tables. In this process, the arithmetic $(D_a)$, geometric $(D_g)$ and harmonic $(D_h)$ forms of calculating the mean value of grain size were used.

### 4.1.1 Sandy aquifer

The hydraulic conductivities of samples from various depths are presented for four distinctive aquifers.

First, two aquifers are built of uniform, poorly graded mean to coarse grained sand (fig. 6) lying on different depths. Second, two aquifers are built of well graded fine to mean grained sand (fig. 7), also lying on different depths.

Table 1. Average difference (%) between predicted and tested hydraulic conductivity for sandy aquifers

Figure 6. Predicted hydraulic conductivity calculated using KC equation for samples from uniform sandy aquifer $(K(D_{40})$ – K calculated using effective grain size $D_{40}$, $K(D_a)$- K calculated using arithmetic mean grain size, $K(D_h)$ - K calculated using harmonic mean grain size, $K(D_g)$ - K calculated using geometric mean grain size)

Figure 7. Predicted hydraulic conductivity calculated using KC equation for samples from sandy aquifers with thin silty intercalations

Table 1 gives the average difference between the predicted and tested (pumping test) hydraulic conductivities. In all cases, the overestimated value of hydraulic conductivity is a result of using the arithmetic mean grain size in calculations. The underestimated values of hydraulic conductivity are a result of using the harmonic mean grain size. The results are very close to tested value of hydraulic conductivity because the geometric mean grain size was used in the KC formula. The applicability of grain sizes according to the specific sieve size was also analyzed for median grain size value $D_{50}$ and smaller grain sizes. Using the median grain size value $(D_{50})$ resulted in the regular overestimation of hydraulic conductivity, and using grain size $D_{30}$ regularly underestimated hydraulic conductivity (Table 1). An especially interesting fact is that the use of grain size $D_{40}$ (Table 1, Fig. (6)) provided remarkable results with practically negligible errors.

The analyses of samples from fine sandy aquifers with silty laminas (Fig. (7), Fig. (8)) resulted in regularly underestimated K-values. The laminas of silt were so thin that it was not possible to isolate the sand content in the samples (Fig. (8)).

Figure 8. Fine sand sample with thin silty intercalations - test field FS/SU1(Ravnik)

In such specific cases, grain size $D_{40}$ or even $D_{50}$ present hydraulic properties of sandy deposits much better than the calculated mean grain size of the whole sample. Thin laminas of silt, through which the horizontal flow is negligible, have a strong impact on the grain size distribution curve. Yet, these distortions are considerably weaker if the referential geometric mean grain size, $D_g$ and not $D_a$ or $D_h$ is used in the calculations.



### 4.1.2 Gravelly aquifer

The predicted K-values of the gravelly aquifer were analyzed through the same procedures as those of the sandy aquifer. Due to clarity, only K-values based on $D_g$, $D_a$, $D_h$ and $D_{40}$ (Table 2, Fig. (9)) are presented. The extreme graduation of deposits is specific to this pilot field. These deposits contain pebbles (of diameters up to 10 cm), sand and small amount of silt (uniformity $U = D_{60}/D_{10} = 17 – 262$).

Figure 9. Gravel core from 23 to 30 m depth from borehole SPB-3 – test field GW (Đurđevac) (see fig. 10a)

A high-quality drilling core (Fig. 9) from six exploration boreholes and a particle size distribution data analysis of relevant core samples was used. All of the boreholes were scattered around the pumped well at test field GW. Borehole SPB-2 is situated on the border of the well field where a part of an aquifer of sandy development is located, and hence, the data do not correspond to a correlated average K-value. The predicted K-values of particular samples and two boreholes (SPB-3, SPB-5) mean values are presented graphically in Fig. (10). The mean predicted $K(D_g)$ of borehole SPB-3 (Fig. 10a) is only 10% smaller than the tested value. The core quality of this borehole is presented by a core segment of depth from 23.0 m to 30.0 m (Fig. (9)).

Figure 10. Predicted hydraulic conductivity calculated using KC equation for samples from gravely aquifer (test field GW) – a) borehole SPB-3; b) borehole SP B-5

The highest deviation of the predicted $K(D_g)$ in relation to the tested $K_t$ value was noted in the borehole SPB-5 core. The average $K(D_g)$ value is 71% higher than $K_t$ value. However, the most important fact is that the geometric mean $K(D_g)$ of all boreholes (Table 2) in the tested area is only 5% higher than $K_t$. Both values are of the same regional significance. Namely, $K(D_g)$ presents 1) the result of total geometric mean size of all of the grains in the sample, 2) the hydraulic conductivity of all of the samples in the borehole and 3) all of the boreholes on the test field. The tested hydraulic conductivity $K_t$ is identified by analyzing the series of successive cones of depression achieved in that area during the long term pumping test. Conversely, $K(D_a)$ shows higher values by two orders of magnitude and $K(D_h)$ shows lower values by three orders of magnitude. This shows the degeneration of arithmetic algorithm for calculating mean grain size for a wide range of particle sizes.

Table 2. Average predicted hydraulic conductivities $K$ (m/s) for boreholes in gravely aquifer (test field GW)

Table 3. Numerical results of correlations between tested $K_t$ and predicted $K$ for samples from test fields in Croatia. and U.S. Geol. Survey laboratory

The correlation of hydraulic conductivity mean value results for referential grain sizes $D_g$, $D_a$, $D_h$ and $D_{40}$ and the tested mean hydraulic conductivity $K_t$ on all pilot fields is presented graphically in Fig. (11a). It is clear that the values of predicted hydraulic conductivity using the referent grain size $D_g$ closely correlate with the tested ($K_t$) value for all incohesive deposits, regardless of their uniformity. Using $D_a$ and $D_h$ results in the overestimation and the underestimation of hydraulic conductivities, respectively. This distortion significantly depends on the graduation of samples. When the sample is poorly graded, distortion was negligible. In the cases of well graded samples, distortion reaches up to a





few orders of magnitude. A very high Pearson's coefficient of correlation (Fig 11 b, Table 3) confirms
the closeness of tested $K_t$ values and the predicted hydraulic conductivity $K(D_g)$.
Figure 11. Graphical correlation between predicted $K$ and tested $K_t$ for sandy and gravely aquifers. (a)
Difference between arithmetic, geometric and harmonic mean grain size, (b) Results of correlation
between predicted $K(D_g)$ and tested $K_t$
From a practical point of view, an interesting fact is that very good results are achieved using
grain size $D_{40}$ (Fig. 11a).

## 4.2 Cohesive deposit

The validities of the aquitard's predicted K-values was analyzed for 86 samples using the
geometric ($D_g$), arithmetic ($D_a$) and harmonic ($D_h$) mean grain sizes. The results of the correlation
between the predicted and laboratory tested hydraulic conductivities for the samples of cohesive
deposits are presented in Fig. (12a). The permeability test and grain size analysis were performed for
each individual sample. The samples were of various compounds of silty and clayey materials, and
their tested hydraulic conductivities have a wide range, exceeding three orders of magnitude (between
$10^{-11}$ and $10^{-7}$ m/s). This wide range ensures reliable graphical and numerical correlations. These
results are similar to the results of previously explained analyses of non-cohesive deposits. The
arithmetic mean grain sizes result in overestimating $K(D_a)$, and the harmonic mean grain sizes result in
underestimating $K(D_h)$ (that is, average $K(D_a)/K_t$ equaled 14.5 and $K(D_h)/K_t$ equaled 0.17). Good
results were achieved using the referential geometrical mean grain size, and the predicted values of
hydraulic conductivity $K(D_g)$ were very close to the tested value $K_t$ (within the set limits of the
accuracy criteria).
Figure 12. Graphical correlation between predicted $K$ and tested $K_t$ for silt and clay deposit. (a)
Difference between arithmetic, geometric and harmonic mean grain size, (b) Result of correlation
between predicted $K(D_g)$ and tested $K_t$
The graphical correlation (Fig. (12b)) illustrates concentrated $K(D_g)$ values in the neighborhood
of the tested value $K_t$, and most of the results are within the range $1/3K_t<K(D_g)<3K_t$. The numerical
correlation confirms their high correlativity, $R^2=0.696$. This is a very high value, especially
considering the fact that some of deviations may be the result of an error in conducting the laboratory
permeability test. The achieved results confirm earlier conclusions that the total geometric mean grain
diameter $D_g$ truly represents the referent mean grain size of the silty-clayey deposits. Additionally, it
was used as a reliable reference point for the verification of the porosity curve $n_e=f(D_g)$, presented in
Fig. (5).

## 5 Discussion

The Kozeny–Carman equation was limited to only calculating the hydraulic conductivity of
incohesive materials (Kasenow 1997), (Kasenow 2010). Additionally, the use of the KC equation for
calculating the hydraulic conductivities of cohesive materials using particle size has been frequently





disputed in numerous papers and reports. The reasons include varied particle size, high proportions of
fine fractions in deposits (Young and Mulligan 2004), electrochemical reaction between the soil
particles and water and large content of particles such as mica (Carrier 2003). All of these factors also
affect the effective porosity, and some of them also affect the mean grain size. Is the effect of the fore-
mentioned factors incorporated (and/or how much) in the size and distribution of effective porosities
and referential mean grain sizes?
Figure 13. Relation between of effects of mean grain size $D_a$, $D_g$ and $D_h$ on predicted hydraulic
conductivity for all analyzed samples
11       The conducted analyses, as graphically summarized in Fig. 13, confirmed that the use of 1)
geometric mean as a referent mean grain size (Eq. 12 or 13) and 2) effective porosity according to Fig.
(5) in the Kozeny–Carman equation forms a model of flow through the porous media. This model is
valid for various soil materials and mixtures with a wide range of hydraulic conductivity values (from
$10^{-12}$ m/s up to $10^{-2}$ m/s). The use of the arithmetic mean $D_a$ and the harmonic mean $D_h$ result in the
overestimation and the underestimation, respectively, of the value of hydraulic conductivity. The
overestimated porosity is followed by the overestimated value of hydraulic conductivity. This can
have a huge impact on predicting the hydraulic conductivity of clayey-silty deposits, which are of very
high total porosity but very low effective porosity. Therefore, the use of total instead of effective
porosity in Eq (14) can lead to a misunderstanding regarding the validity of the harmonic mean grain
size for calculating the hydraulic conductivities of cohesive materials.
22       Pearson's correlation analysis was conducted for the numerical and logarithmic values of
predicted hydraulic conductivities $K(D_g)$ of all of the samples, grouped in three basic data groups
(Table 3). These include non-cohesive materials (gravel and sand), cohesive materials (silt and clay),
and the group of all of the analyzed samples. The verification of the results for the non-cohesive
materials group was conducted for eight more samples from the USGS laboratory (Morris and Johnson
1967). The verification of the results for cohesive materials was conducted by the analyses of two
more samples from the USGS laboratory. The correlation results of all of the $K(D_g)$ are presented in
Fig. (14).
Figure 14. Verification of graphical and numerical correlation between the tested $K_t$ and the predicted
hydraulic conductivity $K(D_g)$ using referential geometric mean size for all samples
34       A separate sub-group was formed by the non-cohesive material data from all five CRO test
fields by using the referent grain size $D_{40}$. This correlation results in very high correlation coefficients.
The lowest values of the correlation coefficients were observed for the silty-clayey materials group,
but their values (in Table 3) certainly confirm the validity of the observed relations. It is very
important to note that the test data used in this research refer to standard, serial tests and that specific
tests may potentially result in even stronger correlations.
40       The graphical correlation between the tested and the predicted hydraulic conductivities (Fig.
(14)) illustrates the universality of the KC model (when applying referential mean grain size $D_g$ and an
effective porosity $n_e$) in a wide range of flow conditions. The very high values of correlation
coefficients $R^2$ (Table. 3) confirm the relations in continuous porous media conditions on a laboratory
scale.



# 6 Conclusions

The following conclusions can be drawn from this study:

1. The geometric mean size of all particles contained in the sample $D_g$ unambiguously affects the permeability and specific surface area of cohesive and non-cohesive deposits, regardless of the grain size and distribution of specific particles. Hence, $D_g$ represents the referential grain size of the sample.

2. The distribution of effective porosities in functions of the referential grain size $n_e = f(D_g)$ is presented graphically for all types of clastic deposits. The graph was constructed following previously reported data and was calibrated according to the congruence between the tested hydraulic conductivity and its predicted value calculated by applying the Kozeny-Carman equation. Thus, this effective porosity presents the flow porosity and is slightly lower than the specific yield commonly referred to the literature.

3. The successful application of the KC flow model confirms its validity in a range of hydraulic conductivities between $10^{-12}$ and $10^{-2}$ m/s. Simultaneously, the value of effective porosity and its relative referential grain size $D_g$ in a range of 1.5 µm to 6 mm has been verified. It can be concluded that, through the presented parameters, the range of applying the Kozeny-Carman model for calculating permeability and specific surface area is extended up to the limits of Darcy's law validity.

4. The value of the referent mean grain size in cases of analyzed non-cohesive samples is very close to the value of the grain size $D_{40}$ (read from grain size distribution curve).

Acknowledgments:

The authors would like to thank Ms. Željka Brkić, Ph.D, Mr. Željko Miklin and Ms. Ivana Žunić Vrbanek for their perseverance and help in collecting large amounts of laboratory data used in this study. This study was supported by the Ministry of Science, Education and Sports of the Republic of Croatia (Basic Hydrogeological Map of the Republic of Croatia 1:100.000 - basic scientific project of Croatian Geological Survey)



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




1 Table 1. Average difference (%) between predicted and tested hydraulic conductivity for sandy
2 aquifers

| | Variety of equivalent grain size | Diameter form grain-size distribution curves | | | Mean grain size | | | Tested $K_t$ (m/s) | Kind of sand |
|---|---|---|---|---|---|---|---|---|---|
| | | $K(D_{30})$ | $K(D_{40})$ | $K(D_{50})$ | $K(D_a)$ | $K(D_h)$ | $K(D_g)$ | | |
| Well fields | SU-1 | -16,5 | -0,1 | +14,3 | +48,5 | -9,1 | +15,8 | $2{,}55*10^{-4}$ | Medium |
| | SU-2 | -37,1 | -1,4 | +32,9 | +48,7 | -13,6 | +9,9 | $2{,}78*10^{-4}$ | uniform |
| | FS/SU-1 | -23,5 | +1,5 | +26,3 | +48,3 | -76,0 | -21,1 | $1{,}16*10^{-4}$ | Fine to |
| | FS/SU-2 | -48,8 | -27,3 | -4,9 | +38,3 | -48,9 | -12,8 | $1{,}40*10^{-4}$ | medium |
| | Average | -31,5 | -6,8 | +17,2 | +46,0 | -36,9 | -2,1 | | |





1   Table 2. Average predicted hydraulic conductivity K (m/s) for boreholes in gravely aquifer (test field
2   GW)

| Bore-hole | $K(D_g)$ | | $K(D_a)$ | | $K(D_h)$ | | $K(D_{40})$ | | Tested $K_t$ (m/s) |
|---|---|---|---|---|---|---|---|---|---|
| | Geom. | Aritm. | Geom. | Aritm. | Geom. | Aritm. | Geom. | Aritm. | |
| SPB-1 | 2,5E-03 | 3,5E-03 | 5,5E-02 | 5,8E-02 | 6,6E-06 | 8,7E-06 | 1,1E-03 | 2,4E-03 | |
| SPB-3 | 1,6E-03 | 2,5E-03 | 5,9E-02 | 6,4E-02 | 2,2E-06 | 3,3E-06 | 6,4E-04 | 1,6E-03 | |
| SPB-4 | 1,3E-03 | 2,2E-03 | 4,3E-02 | 4,9E-02 | 1,4E-06 | 1,8E-06 | 5,1E-04 | 1,1E-03 | 1,8E-03 |
| SPB-5 | 3,0E-03 | 4,2E-03 | 5,5E+02 | 5,6E-02 | 5,7E-06 | 8,3E-06 | 1,6E-03 | 4,6E-03 | |
| SPB-6 | 1,2E-03 | 1,4E-03 | 2,6E-02 | 2,8E-02 | 2,2E-06 | 2,4E-06 | 7,1E-04 | 8,8E-04 | |
| Aver. | 1,8E-03 | 2,6E-03 | 2,9E-01 | 4,9E-02 | 3,1E-06 | 4,0E-06 | 8,4E-04 | 1,8E-03 | |
| $K/K_t$ | 1,02 | 1,47 | 163 | 28 | 0,0017 | 0,0023 | 0,48 | 1,01 | |





1  Table 3. Numerical results of correlations between tested $K_t$ and predicted $K$ for samples from test
2  fields in Croatia. and U.S. Geol. Survey laboratory

| Samples from | Materials | Referential mean grain size | Mark | Pearson's correlation coeffecients | | | |
| --- | --- | --- | --- | --- | --- | --- | --- |
| | | | | Nominal values | | Log values | |
| | | | | R | $R^2$ | R | $R^2$ |
| CRO test fileds | Gravel, sand | $D_g$ | $R_1$ | 0,999 | 0,998 | 0,988 | 0,976 |
| | Gravel, sand | $D_{40}$ | $R_2$ | 1,000 | 1,000 | 0,995 | 0,991 |
| Togeather CRO + USGS lab. | Gravel, sand | $D_g$ | $R_3$ | 0,997 | 0,994 | 0,993 | 0,985 |
| CRO test fileds | Silt, clay | $D_g$ | $R_4$ | 0,740 | 0,547 | 0,834 | 0,696 |
| | Gravel, sand, silt,clay | $D_g$ | $R_5$ | 1,000 | 0,999 | 0,971 | 0,942 |
| All togeather CRO+USGS lab. | Gravel, sand, silt,clay | $D_g$ | $R_6$ | 0,997 | 0,995 | 0,985 | 0,971 |





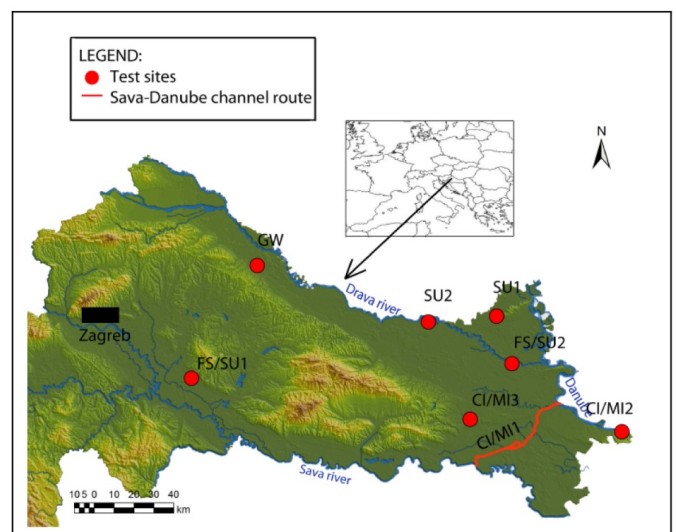

2      Figure 1. The map of Northern Croatia with test sites locations



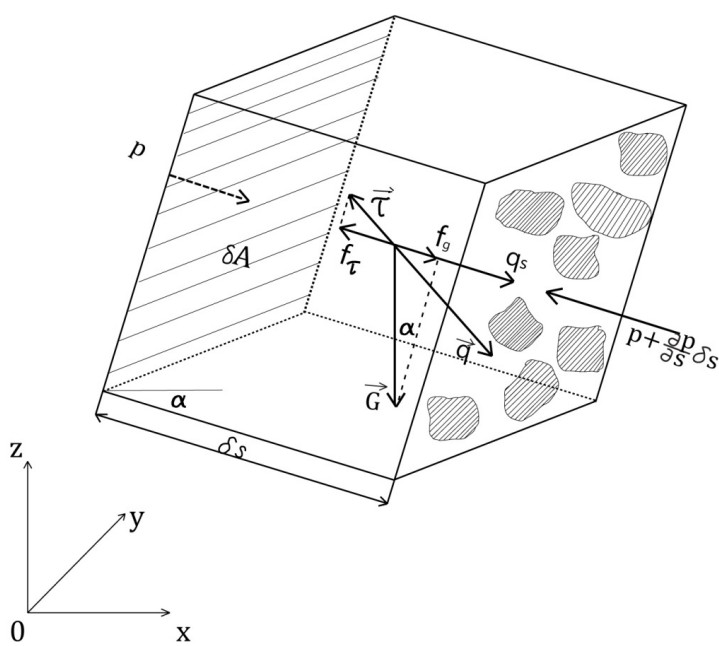

2    Figure 2. Definition sketch of liquid driving and opposed viscous forces for elemental volume





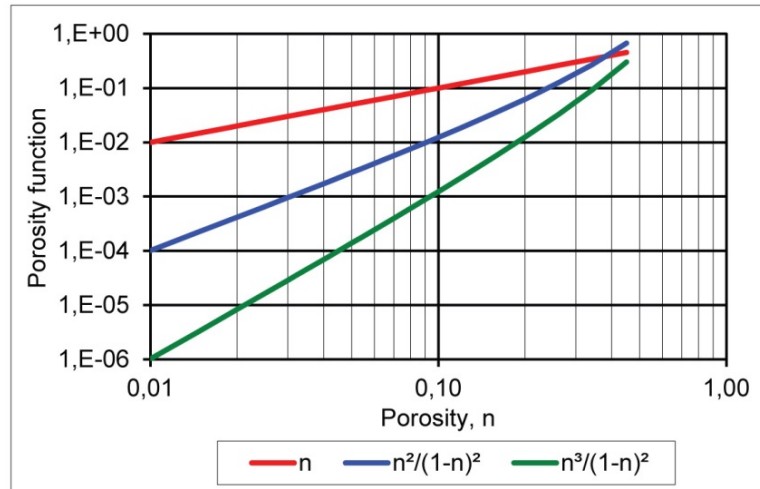

2      Figure 3. Effects of driving (n) and drag resistance ($n^2/(1-n)^2$) factors in porosity function ($n^3/(1-n)^2$)



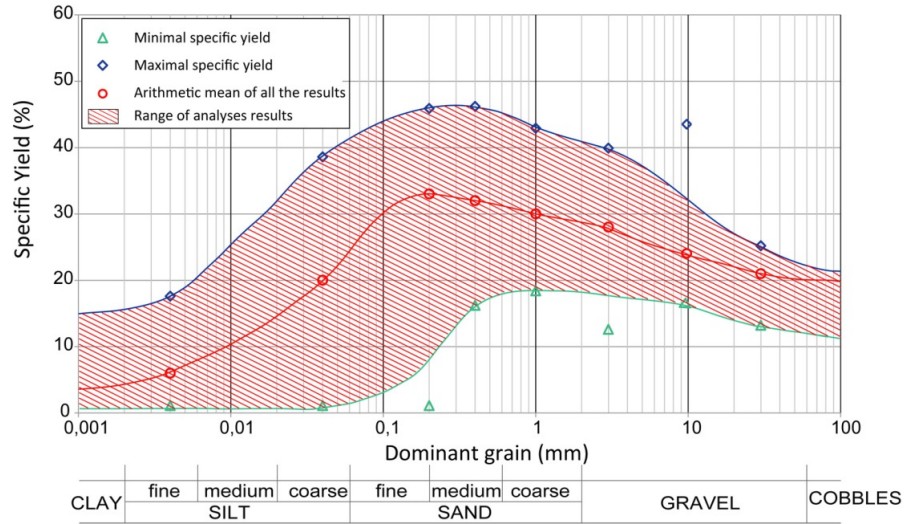

2    Figure 4. Range and arithmetic mean of specific yield values for 586 analyses in Hydrol. Lab. of the

3    U.S. Geol. Survey (from Morris & Johnson, 1967)



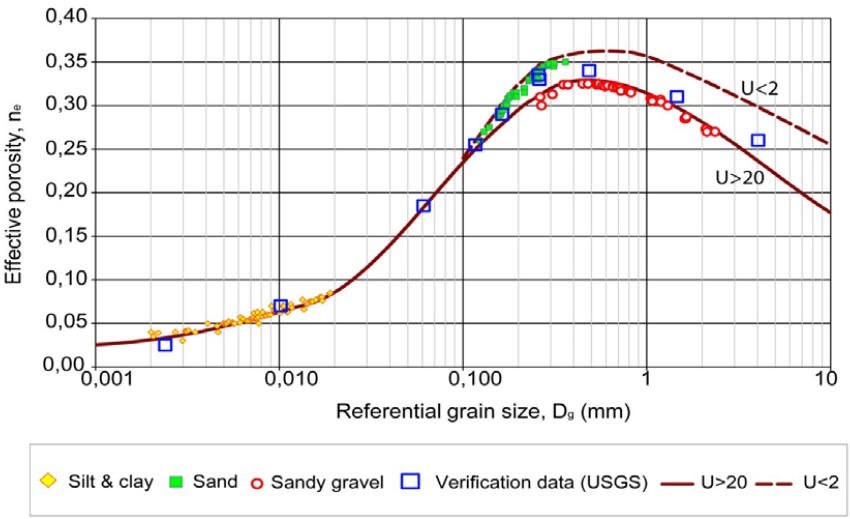

2    Figure 5. Relation between referential mean grain Dg and effective porosity $n_e$. Note: Dot line divides

3    uniform grain deposits U=D60/D10<2, and medium uniform grain deposit 2<U<20. Verified samples

4    of non-uniform grain deposits of sand and gravel (U>20) lie below the full line



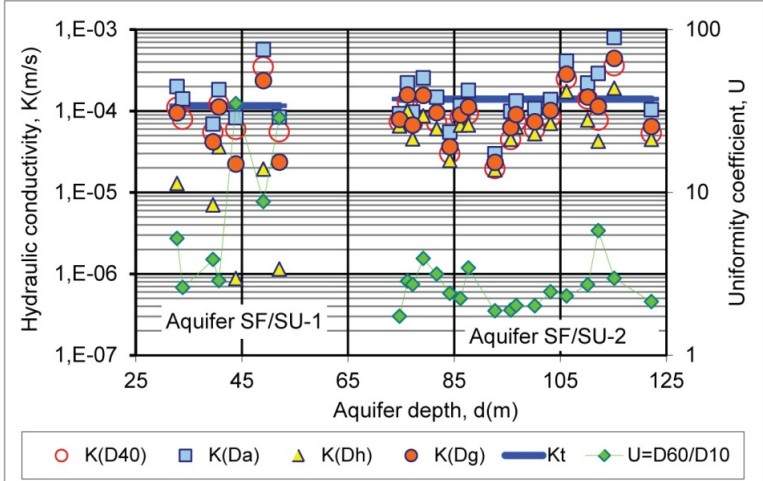

Figure 6. Predicted hydraulic conductivity calculated using KC equation for samples from uniform
sandy aquifer ($K(D_{40})$ – K calculated using effective grain size $D_{40}$, $K(D_a)$ - K calculated using
arithmetic mean grain size, $K(D_h)$ - K calculated using harmonic mean grain size, $K(D_g)$ - K calculated
using geometric mean grain size)





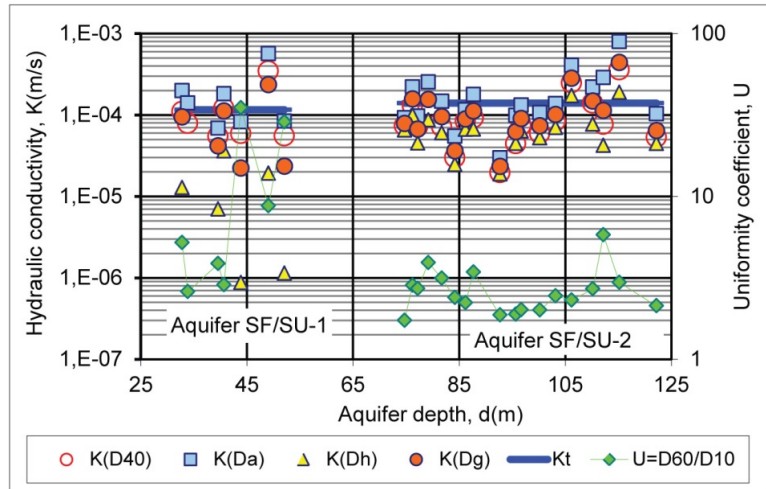

2    Figure 7. Predicted hydraulic conductivity calculated using KC equation for samples from sandy

3    aquifers with thin silty intercalations



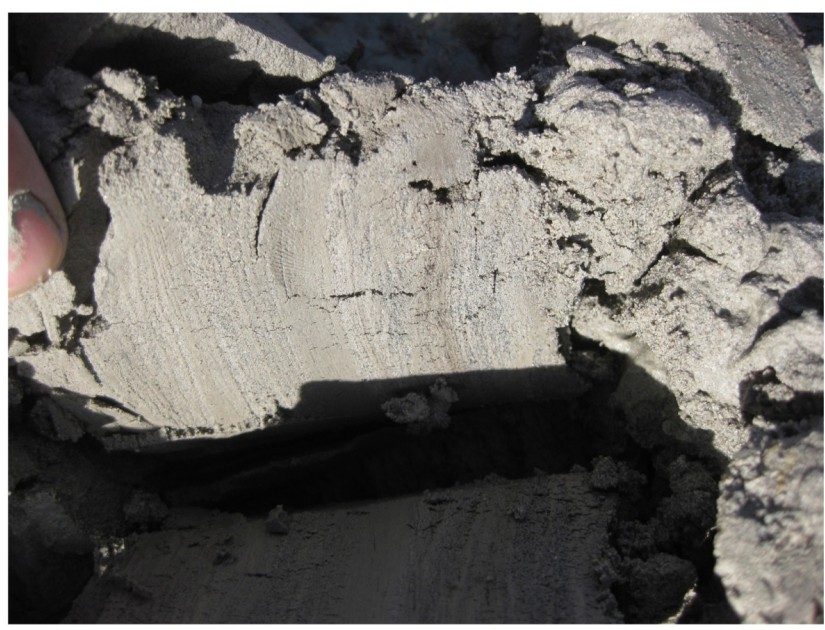

2     Figure 8. Fine sand sample with thin silty intercalations - test field FS/SU1 (Ravnik)





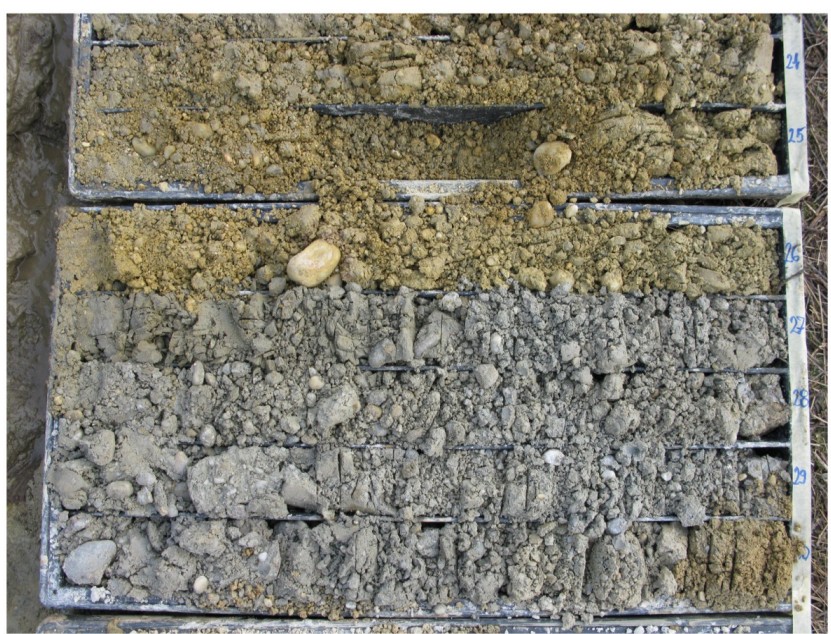

2    Figure 9. Gravel core from 23 to 30 m depth from borehole SPB-3 – test field GW (Đurđevac) (see
3    fig. 10a)



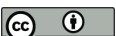

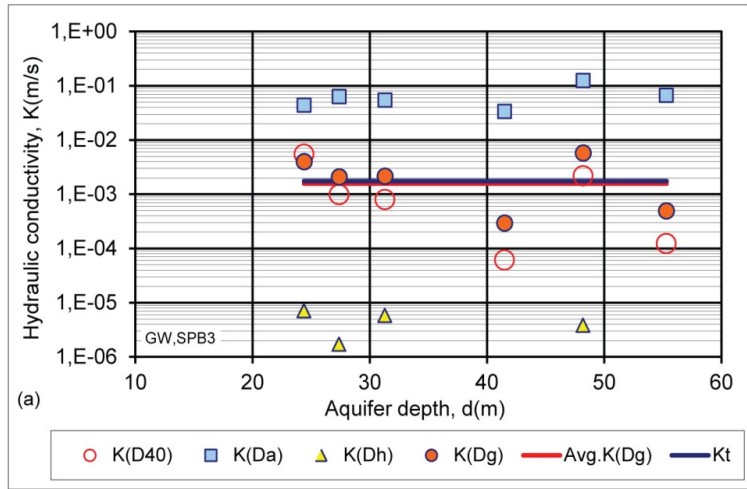

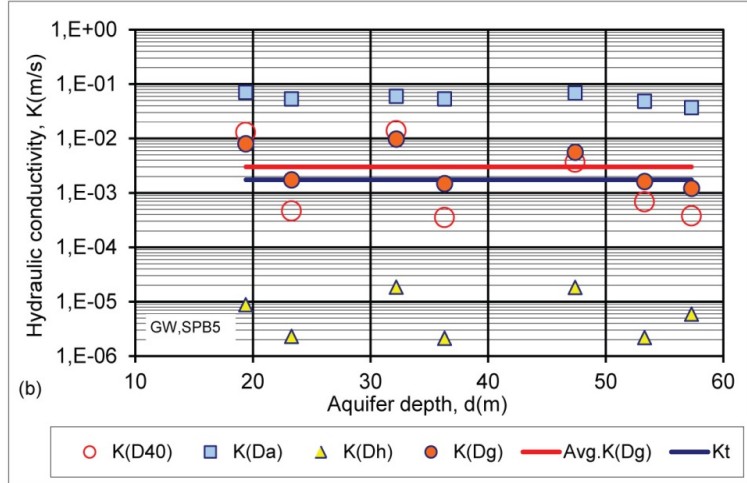

1     ,

2  Figure 10. Predicted hydraulic conductivity calculated using KC equation for samples from gravely

3  aquifer (test field GW) – a) borehole SPB-3; b) borehole SP B-5




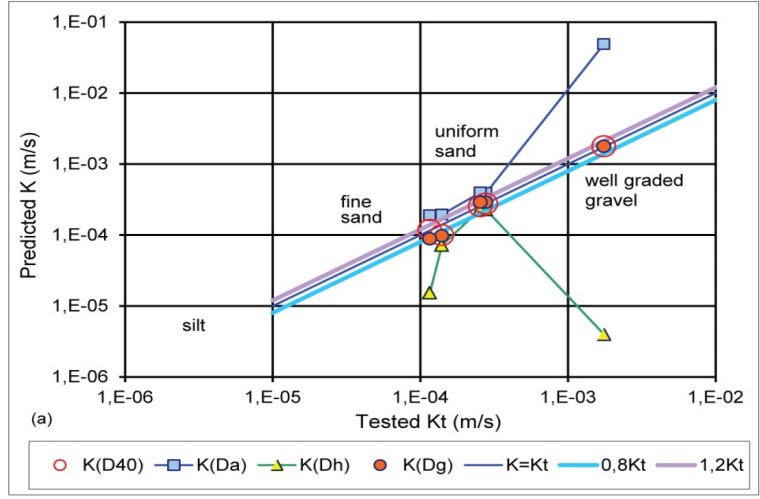

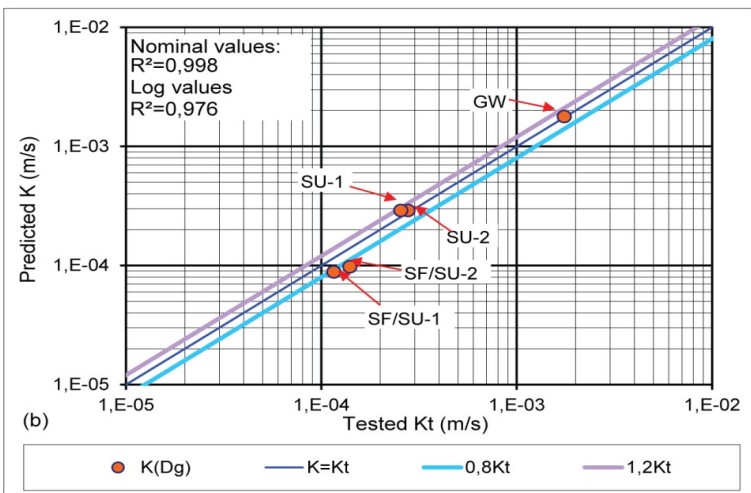

2    Figure 11. Graphical correlation between predicted $K$ and tested $K_t$ for sandy and gravely aquifers. (a)

3    Difference between arithmetic, geometric and harmonic mean grain size, (b) Results of correlation

4    between predicted $K(D_g)$ and tested $K_t$





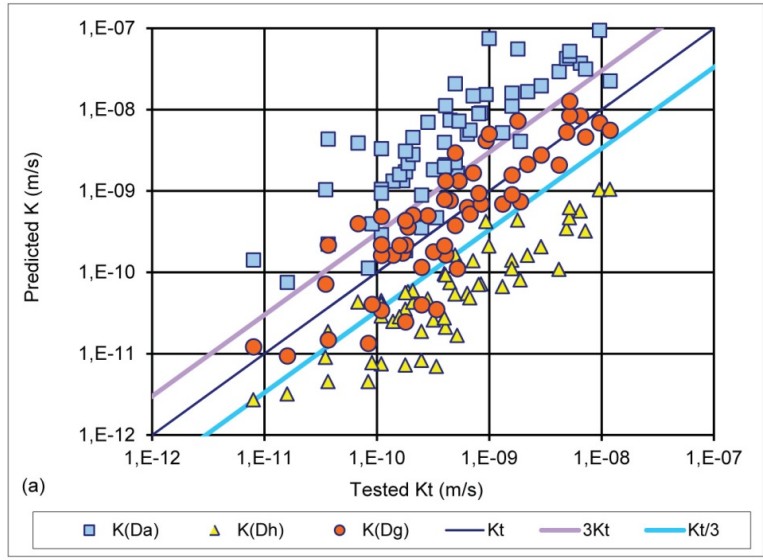

(a)

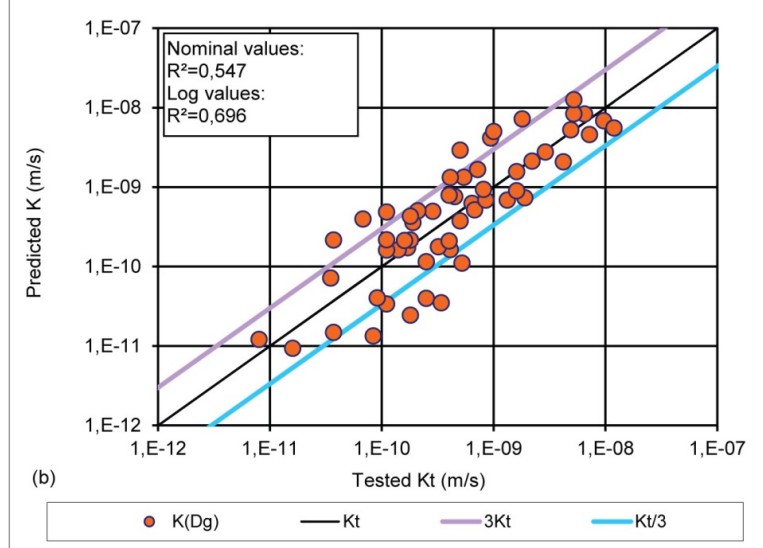

(b)

2 Figure 12. Graphical correlation between predicted $K$ and tested $K_t$ for silt and clay deposits. (a)

3 Difference between arithmetic, geometric and harmonic mean grain size, (b) Result of correlation

4 between predicted $K(D_g)$ and tested $K_t$





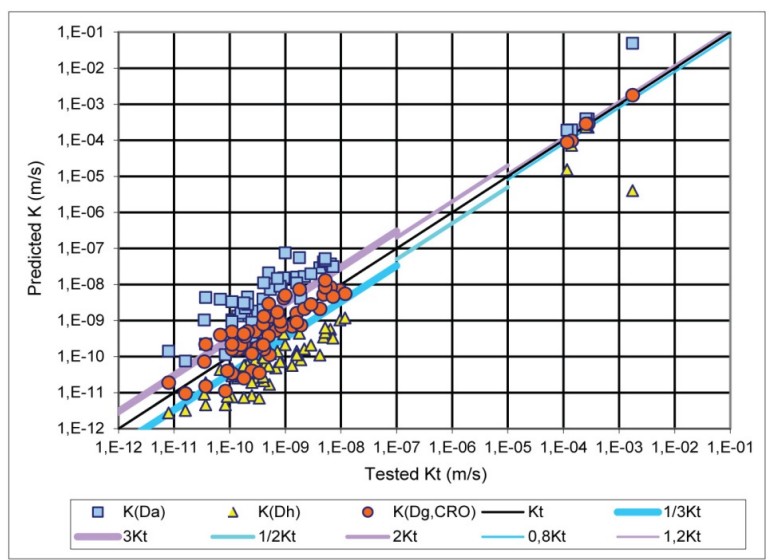

Figure 13. Relation between of effects of mean grain size $D_a$, $D_g$ and $D_h$ on predicted hydraulic
conductivity for all analyzed samples





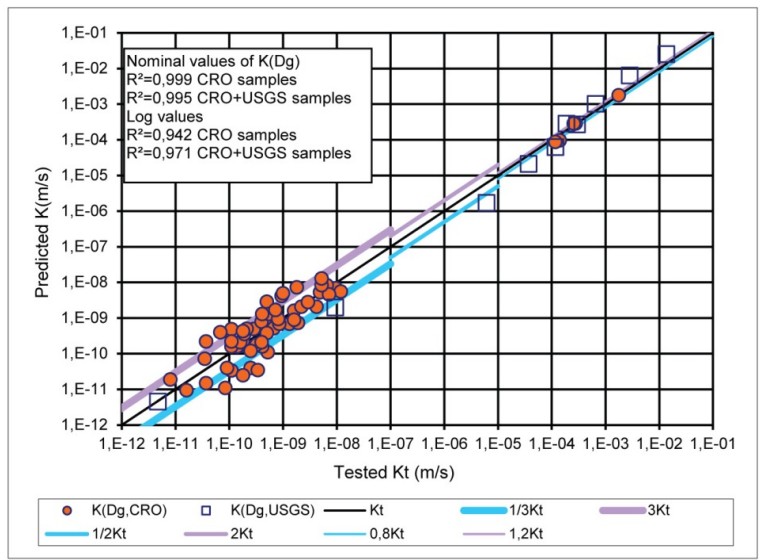

2    Figure 14. Verification of graphical and numerical correlation between the tested $K_t$ and the predicted

3    hydraulic conductivity $K(D_g)$ using referential geometric mean size for all samples

