# Peer review of "The referential grain size and effective porosity in the Kozeny-Carman model"

_Hydrology and Earth System Sciences, 2015_

## Referee Comment (RC1) · Anonymous Referee #1 · 21 Jan 2016

General comments: Determination of soils permeability on the basis of their grains size is definitely very interesting because it allows to predict the permeability from easily measured and routinely obtainable data. The article has a great value also because the investigations were carried out on a large number of samples. Although I do not agree with all conclusions I think that the article is worthy for publication.

Specific comments: The question is if Kozeny-Carman equation also applies to clays or sands with a larger amount of clay minerals. For example: the studies of Carman (1939) have shown that the KC equation is suitable for the evaluation of permeability for gravel and sand, whereas it is useless for clays. Such a conclusion was based on the studies of natural clays, which showed that the relationship between k and is not constant but decreasing function of porosity. Experimental investigations of Taylor (1948) have confirmed this claim, as well as measurements of fine grained natural

materials, carried out by Michaels and Lin (1954). Al-Tabbaa and Wood (1987) have demonstrated that the coefficient of permeability for kaolinite is not linearly dependent on , which means that the KC equation does not apply. By the same conclusion also came Dolinar and Otoničar, 2007. They used pure clay minerals in their investigations. They concluded that KC equation is not suitable for clays in original form. They proposed a modified form of KC equation (Geologija, 2007, vol. 50, No. 2, str. 487-495). There is also the question how to properly measure the grain size of the fine-grained soils. With the use of hydometer method, which is commonly used method for engineering purposes, the results are not precise enough. It is well known that very small amount of clay minerals have a great influence to the permeability of soils. I believe that the assessment of the permeability of cohesive soils is, in the manner suggested by the authors, less reliable, while it is very good for non-cohesive soils.

Technical corrections: Page 4, line 17: specific surface area based on the mass of solids Ms Page 4, line 31: . . . are efective porosity ne (not n)
* * *

---

## Author Comment (AC1) · 4 Feb 2016

RC:General comments: Determination of soils permeability on the basis of their grains size is definitely very interesting because it allows to predict the permeability from easily measured and routinely obtainable data. The article has a great value also because the investigations were carried out on a large number of samples. Although I do not agree with all conclusions I think that the article is worthy for publication.

AUTHOR'S ANSWER: We thank the reviewer for swift and prompt review. Also, we are glad to hear that reviewer 1 agrees with our opinion that this is an interesting and useful topic.

Specific comments: The question is if Kozeny-Carman equation also applies to clays or sands with a larger amount of clay minerals. For example: the studies of Carman

(1939) have shown that the KC equation is suitable for the evaluation of permeability for gravel and sand, whereas it is useless for clays. Such a conclusion was based on the studies of natural clays, which showed that the relationship between k and is not constant but decreasing function of porosity. Experimental investigations of Taylor (1948) have confirmed this claim, as well as measurements of fine grained natural materials, carried out by Michaels and Lin (1954). Al-Tabbaa and Wood (1987) have demonstrated that the coefficient of permeability for kaolinite is not linearly dependent on , which means that the KC equation does not apply. By the same conclusion also came Dolinar and Otonicar, 2007. They used pure clay minerals in their investigations. They concluded that KC equation is not suitable for clays in original form. They proposed a modified form of KC equation (Geologija, 2007, vol. 50, No. 2, str. 487-495). There is also the question how to properly measure the grain size of the fine-grained soils. With the use of hydometer method, which is commonly used method for engineering purposes, the results are not precise enough. It is well known that very small amount of clay minerals have a great influence to the permeability of soils. I believe that the assessment of the permeability of cohesive soils is, in the manner suggested by the authors, less reliable, while it is very good for non-cohesive soils.

AUTHOR'S ANSWER: This manuscript indicates that Kozeny-Carman model is suitable for calculating hydraulic conductivity within the limits of validity of Darcy's law. In both historic and recent scientific literature it was stated that Kozeny-Carman formula is only suitable for evaluation of hydraulic conductivity of gravel and sand. We do agree with this evaluation within the up to now-limitations of factors in KC formula. My impulse to thoroughly study this method was the fact that KC method is completely logical and theoretically correct. Therefore there must have been a way to apply it on natural sediments of various granulometric compounds. We have tried to optimize factors in KC equation. Then, while studying the porosity, we have come to the conclusion that real effective (flow) porosity is not the same as recently used specific yield. There is a small difference between two mentioned forms. Real effective porosity is associated with liquid flow velocity (relations of Darcy's and Hagen-Poiseuill's velocity) and therefore

presents a property of saturated media. Specific yield is a property describing desaturation of an aquifer, and is therefore time dependent of time. The other factor that was optimized was referential grain size. The idea to use geometric mean grain size was described in the manuscript (page 5, lines 27-35 and 6, lines 2-6). We believe that this optimization of factors in Kozeny-Carman formula led to a significant expanding of granulometric range that formula can be applied on. Range of applicability of KC formula was expanded on fine grained sediments up to referential grain size 0,003 mm. To summarize this thesis, the effect of change of porosity was expressed through porosity function, and value of porosity in relation with referential grain size was presented graphically in Figure 5. That, we believe, was the main scientific contribution of this manuscript. The studied samples of silty clay were undisturbed samples of natural deposits, from borehole core where quartz mineral was dominant. The impact of particular clay particles was not analyzed. It is beyond doubt that mineral composition of samples has a strong impact on hydraulic conductivity. And that is probably the reason why the correlation coefficient for cohesive (clayey) deposits is significantly lower than correlation coefficient of non-cohesive deposits. Further development of these research was planned in order to answer the above questions.

Technical corrections: Page 4, line 17: specific surface area based on the mass of solids Ms - OK, corrected Page 4, line 31: are efective porosity ne (not n) – OK, corrected

---

## Referee Comment (RC2) · Anonymous Referee #2 · 17 Feb 2016

This paper explores an interesting subject, showing the behaviour of porous media with grain sizes varying from silty clay to gravels, comparing the results from pumping tests and the KC equation for samples taken at different depths.

I believe that the visual comparison of the values of field and laboratory, shown in the paper, helps demonstrate the arguments outlined by the authors.

I recommend the publication of this paper.

Corrections that should be made: Page 4, line 31 says "...effective porosity n..." and should be "...effective porosity ne..." In figure 6, it is not indicated what does Kt stands for (tested hydraulic conductivity).

---

## Author Comment (AC2) · 7 Mar 2016

RC: This paper explores an interesting subject, showing the behaviour of porous media with grain sizes varying from silty clay to gravels, comparing the results from pumping tests and the KC equation for samples taken at different depths. I believe that the visual comparison of the values of field and laboratory, shown in the paper, helps demonstrate the arguments outlined by the authors. I recommend the publication of this paper

Author's comment: The authors would like to thank the Anonymous Reviewer #2 for her/his review. We are delighted to have received similar opinions from both Reviewers. We believe that this truly is a very interesting and useful research.

RC: Corrections that should be made: Page 4, line 31 says "...effective porosity n..." and should be "...effective porosity ne..." In figure 6, it is not indicated what does Kt

stands for (tested hydraulic conductivity)

Author's comment: Referring to specific corrections, we'll accept both corrections. Thank You for contributing to this paper.

Kind regards and greetings from Croatia!

—————————————————

---

## Author Response (AR2)

**COMMENTS FROM REFEREES:**

**Reviewer 1**

REVIEWER'S COMMENT: General comments: Determination of soils permeability on the basis of their grains size is definitely very interesting because it allows to predict the permeability from easily measured and routinely obtainable data. The article has a great value also because the investigations were carried out on a large number of samples. Although I do not agree with all conclusions I think that the article is worthy for publication.

AUTHOR'S ANSWER: We thank the reviewer for swift and prompt review. Also, we are glad to hear that reviewer 1 agrees with our opinion that this is an interesting and useful topic.

REVIEWER'S COMMENT Specific comments: The question is if Kozeny-Carman equation also applies to clays or sands with a larger amount of clay minerals. For example: the studies of Carman (1939) have shown that the KC equation is suitable for the evaluation of permeability for gravel and sand, whereas it is useless for clays. Such a conclusion was based on the studies of natural clays, which showed that the relationship between k and is not constant but decreasing function of porosity. Experimental investigations of Taylor (1948) have confirmed this claim, as well as measurements of fine grained natural materials, carried out by Michaels and Lin (1954). Al-Tabbaa and Wood (1987) have demonstrated that the coefficient of permeability for kaolinite is not linearly dependent on , which means that the KC equation does not apply. By the same conclusion also came Dolinar and Otonicar, 2007. They used pure clay minerals in their investigations. They concluded that KC equation is not suitable for clays in original form. They proposed a modified form of KC equation (Geologija, 2007, vol. 50, No. 2, str. 487-495). There is also the question how to properly measure the grain size of the fine-grained soils. With the use of hydometer method, which is commonly used method for engineering purposes, the results are not precise enough. It is well known that very small amount of clay minerals have a great influence to the permeability of soils. I believe that the assessment of the permeability of cohesive soils is, in the manner suggested by the authors, less reliable, while it is very good for non-cohesive soils.

AUTHOR'S ANSWER: This manuscript indicates that Kozeny-Carman model is suitable for calculating hydraulic conductivity within the limits of validity of Darcy's law. In both historic and recent scientific literature it was stated that Kozeny-Carman formula is only suitable for evaluation of hydraulic conductivity of gravel and sand. We do agree with this evaluation within the up to now-limitations of factors in KC formula. My impulse to thoroughly study this method was the fact that KC method is completely logical and theoretically correct. Therefore there must have been a way to apply it on natural sediments of various granulometric compounds. We have tried to optimize factors in KC equation. Then, while studying the porosity, we have come to the conclusion that real effective (flow) porosity is not the same as recently used specific yield. There is a small difference between two mentioned forms. Real effective porosity is associated with liquid flow velocity (relations of Darcy's and Hagen-Poiseuill's velocity) and therefore presents a property of saturated media. Specific yield is a property describing desaturation of an aquifer, and is therefore time dependent. The other factor that was optimized was referential grain size. The idea to use geometric mean grain size was described in the manuscript (page 5, lines 27-35 and 6, lines 2-6). We believe that this optimization of factors in Kozeny-Carman formula led to a significant expanding of granulometric range that formula can be applied on. Range of applicability of KC formula was expanded on fine grained sediments up to referential grain size 0,003 mm. To summarize this thesis, the effect of change of porosity was expressed through porosity function, and value of porosity in relation with referential grain size was presented graphically in Figure 5. That, we believe, was the main scientific contribution of this manuscript. The studied samples of silty clay were undisturbed samples of natural deposits, from borehole core where quartz mineral was dominant. The impact of particular clay particles was not analyzed. It is beyond doubt that mineral composition of samples has a strong impact on hydraulic conductivity. And that is probably the reason why the correlation coefficient for cohesive (clayey) deposits is significantly lower than correlation coefficient of non-cohesive deposits. Further development of these researches was planned in order to answer the above questions.

Change in Manuscript - One small paragraph added in Introduction – adding two references (Al Tabbaa, 1987 and Dolinar 2007)

REVIEWER'S COMMENT Technical corrections: Page 4, line 17: specific surface area based on the mass of solids $M_s$ –

AUTHOR'S ANSWER OK, corrected

REVIEWER'S COMMENT Page 4, line 31: are efective porosity $n_e$ (not n) –

AUTHOR'S ANSWER OK, corrected

**Reviewer 2**

REVIEWER'S COMMENT: This paper explores an interesting subject, showing the behaviour of porous media with grain sizes varying from silty clay to gravels, comparing the results from pumping tests and the KC equation for samples taken at different depths. I believe that the visual comparison of the values of field and laboratory, shown in the paper, helps demonstrate the arguments outlined by the authors. I recommend the publication of this paper.

AUTHOR'S ANSWER: The authors would like to thank the Anonymous Reviewer #2 for her/his review. We are delighted to have received similar opinions from both Reviewers. We believe that this truly is a very interesting and useful research.

REVIEWER'S COMMENT: Corrections that should be made: Page 4, line 31 says "...effective porosity n..." and should be "...effective porosity ne..."

AUTHOR'S ANSWER OK, corrected

REVIEWER'S COMMENT In figure 6, it is not indicated what does Kt stands for (tested hydraulic conductivity)

AUTHOR'S ANSWER OK, corrected

LIST OF ALL THE RELEVENT CHANGES:

1.  Page 2; Line 25 - 3 SENTENCES INSERTED IN INTRODUCTION: "Several authors (Al-Tabbaa and Wood, 1987, Dolinar and Otoničar, 2007) have studied applicability of KC formulae for calculation of hydraulic conductivity of fine grained materials. All of them have concluded that KC model in its original form does not apply on clays. Dolinar and Otoničar (2007) have also proposed modified form of KC equation."
2.  Page 4, line 17 – "area" – inserted
3.  Page 4, line 31 - $n_e$ – corrected
4.  Page 9 –line 21 – caption of Figure 6; "Kt – tested hydraulic conductivity)" – inserted
5.  Two references inserted:

[revised manuscript text omitted]